# Qualitative exploration of the intersection between social influences and cultural norms in relation to the development of alcohol use behaviour during adolescence

Georgie J MacArthur ![ORCID], Matthew Hickman ![ORCID], Rona Campbell

Department of Population Health Sciences, Bristol Medical School, University of Bristol, Bristol, UK

**Correspondence to**
Dr Georgie J MacArthur;
georgie.macarthur@bristol.ac.uk

## ABSTRACT

**Objectives** Few contemporary studies have examined peer and social drivers of alcohol use during mid-adolescence. We sought to explore young people's perspectives on socio-cultural influences relating to alcohol use behaviour during this period.

**Design** Qualitative research study.

**Methods** Semi-structured one-to-one (n=25), paired (n=4) or triad (n=1) interviews and one focus group (n=6) were conducted with 30 young people aged 14 to 15 (13 males, 17 females) recruited from 4 schools, and 12 participants (aged 14 to 18, 8 males, 4 females) recruited from two youth groups in an urban centre in the West of England. Nineteen participants abstained from alcohol use, 9 were occasional or moderate drinkers and 14 drank alcohol more regularly. Interviews were audio-recorded, transcribed verbatim and analysed thematically using NVivo V.10, through a lens of social influence and social norms theories.

**Results** Alcohol consumption was associated with being cool, mature and popular, while enabling escape from reality and boosting confidence and enjoyment. Positive expectancies, alongside opportunity, contributed to motivating initiation, but social influences were paramount, with participants describing a need to 'fit in' with friends to avoid social exclusion. Such influences positioned drinking at parties as a normative social practice, providing opportunities for social learning and the strengthening of peer norms. Social media presented young people with positive alcohol-associated depictions of social status, enjoyment and maturity. This intersection of influences and norms generated a pressurised environment and a sense of unease around resisting pressures, which could elicit stigmatising insults.

**Conclusions** Cultural norms, social influences and social media intersect to create a pressurised environment around alcohol use during mid-adolescence, driving the escalation in the prevalence of excessive consumption at this stage. New interventions need to address normative influences to enable the prevention of excessive alcohol use during adolescence.

## INTRODUCTION

Although downward trends in alcohol use have been reported in the USA and Europe,[1 2]

**Strengths and limitations of this study**

► Strengths of the study include the gender-balanced sample of young people from both schools and youth groups, including those who used alcohol and those who abstained.

► One-to-one interviews were conducted, which provided space for young people to share their views around drinking, without concerns around disagreement or judgement from peers or friends.

► Limitations of this study include the lack of triangulation of data, and participants were recruited from urban and suburban settings only, meaning that our findings cannot necessarily be generalised to different geographical or demographic contexts.

the reasons for this decline are as yet unclear, and while evidence is scant, studies indicate that the decline may not be uniform across all socio-demographic or levels of alcohol consumption.[3]

Despite these declines in the UK, recent figures show that 24% of 15 year olds in the UK have had a drink in the last week,[4] and by the age of 17, half of girls, and nearly two-thirds of boys, report drinking every week.[5] Adolescent drinking in the UK therefore remains a public health concern. Guidance from the Chief Medical Officer for England states that an alcohol-free childhood is the healthiest and best option, and if children do drink alcohol, it should not be until at least the age of 15 years; while if those aged 15 to 17 do drink, they should do so infrequently and on no more than 1 day per week.[6]

The critical role of both peer and parental influences in relation to adolescent alcohol consumption is well documented. Quantitative and qualitative studies have highlighted the impact of factors including parental monitoring, attitudes, communication and

parental provision of alcohol via direct and indirect mechanisms.[7–12] In addition, peer influences, peer selection, the nature and reciprocity of friendships and the social context of drinking, together generate peer group-based behavioural drivers of alcohol consumption,[13–21] and both descriptive and injunctive norms have been shown to be positively associated with alcohol use.[22 23] Exposure to friends' alcohol-related content via social media has also been associated with alcohol use and later heavy episodic drinking, via the development of favourable peer injunctive norms,[24] demonstrating the links between use of social media, norms and social influences.

Normative social influence describes the social and emotional motivation to comply with the majority and the expectations of others to gain social approval and/or to avoid ridicule and social rejection, particularly in public settings where lack of conformity may be more evident.[25 26] Perceptions of the consensus view in a group, or collective norms, influence conformity, since the more consensual the group, the more isolated the 'deviant' and the more powerful the group in shaping the social space. In this way, alcohol use behaviour can be influenced by perceived descriptive norms (the perceived prevalence of alcohol consumption) and injunctive norms (the perceived approval of alcohol use) among young people and both have been shown to influence adolescent alcohol use at the individual and group level.[27 28]

While qualitative studies have reported the effect of peer, parental and wider cultural influences on young people's drinking, interaction and engagement with social networking sites (SNS) and the integral nature of the social context,[13–15 29–32] few recent qualitative studies have involved an in-depth examination of social and cultural influences on the initiation and escalation of alcohol use during early-adolescence to mid-adolescence in England, and the interaction between such behavioural influences. We sought to obtain data on social influences that would be applicable to the national context to inform development of preventive interventions. In addition, since little data are available to help to explain downward trends in alcohol use among young people in England,[2 3 33] it also remains important to gain a contemporary understanding of the impact of peers, social influences and cultural factors to offer insights around factors that might affect such trends.

In this paper, we report findings of a qualitative study that used theories of social influences and social norms as the theoretical lens through which to explore the social, cultural and behavioural drivers of alcohol consumption and the nature of drinking culture in mid-adolescence. Our exploration of the social worlds and drinking cultures of younger adolescents, and influences on drinking practice, aims to inform the development of interventions to prevent excessive alcohol consumption and related harm.

## METHODS

While the initial aim of the study was to investigate the role of friendships in relation to drinking behaviour, we report here our findings regarding this and a wider range of determinants of behaviour, which reflects additional topics pertinent to young people that were raised by them and discussed in the interviews and which thereby feature in the data analysis.

### Sampling and recruitment

Our study sample included 30 young people aged 14 to 15 years who were recruited from four secondary schools in the South West of England. No participants dropped out of the study. The schools were in urban (n=3) and suburban/rural (n=1) areas and were in areas with diverse levels of deprivation as reflected by the index of multiple deprivation score for the ward. The index of multiple deprivation score for the ward within which the school was located was identified. Schools were grouped by Index of Multiple Deprivation (IMD) score and a random number generator used to select a school from each group to be contacted regarding participation. The final sample of four schools represented wards with varied ward-level IMD scores. Students were randomly selected from the year group and snowball sampling was employed subsequently to recruit additional participants. A further 12 participants aged 14 to 18 years were recruited from youth groups, via leaders of the youth group, to ensure diversity in the sample in relation to socio-demographic characteristics and to enable data collection outside of the school environment. Leaders of youth groups disseminated information about the study and study materials and arranged interviews or the focus group. The sample included individuals who had, and hadn't, consumed alcohol. Parental consent and participant assent were given prior to participation for those aged under 16. Participants aged over 16 years gave informed consent. Recruitment took place between January 2017 and October 2017. The number of participants recruited was determined by the point at which saturation was reached, that is, when no new themes or perspectives were emerging in the interviews.

### Data collection

A total of 36 semi-structured interviews were conducted (15 males, 21 females), four of which were paired interviews, and one of which was a triad interview, conducted with friends. Interviews were conducted in meeting rooms in schools (n=30) and youth groups (n=6). One focus group (n=6 males) was also conducted in a youth group. Interviews were conducted by GJM (PhD), a female researcher with training in qualitative research. No relationship was established with study participants before commencement of data collection. Before starting the interview, participants were aware of the reasons for doing the research and the interviewer's aim of developing a preventive intervention in the future. Interviews were facilitated by a topic guide, which was used flexibly so that participants could explore views and opinions which

were meaningful to them. The topic guide included questions around attitudes and perspectives around drinking, personal histories of alcohol use, risks and consequences of drinking, the role and influence of friendship groups and social networks and family-related factors. Views on alcohol-related education and alcohol-related interventions were discussed but will be reported elsewhere. The participant sample included those who consumed alcohol and those who abstained, but many of those who chose not to drink alcohol discussed their views around alcohol use and perceptions and experiences relating to peers who did consume alcohol. Drinking patterns of participants are noted alongside quotes, based on comments made by participants during interviews. Interviews lasted on average 39 min (range 19 to 66 min) and were audio-recorded and transcribed verbatim. Interview data were anonymised. Participants received a £15 gift voucher for taking part.

## Data analysis

Transcripts of interviews were imported into NVivo V.10 (QSR International, Brisbane) and analysed thematically using this software by GJM. Participants did not check or correct transcripts. Analysis aimed to capture emergent concepts and thus took an inductive approach. Transcripts were read and re-read and open line-by-line coding was conducted with a focus on understanding and capturing participants' experiences, behaviours, feelings, attitudes and perceptions around alcohol use. We sought to explore relationships and patterns both within and between accounts. Different accounts were compared and contrasted, such that codes were progressively refined, and connections mapped, to characterise relationships and identify underlying concepts and then categories and themes. Ongoing engagement with the data ensured that emerging concepts were grounded in participants' accounts. Throughout the process the researcher (GJM) wrote analytical memos and notes to capture thoughts around meanings, themes and relationships between themes. Emerging themes and concepts, and the theoretical basis for analysis, was discussed with the last author. Thematic patterns were considered within and across cases and between groups of cases for example, those who did, or did not, drink alcohol.

Sociological theories relating to social influences, group identity and social norms were explored during analysis to enhance understanding around the theoretical basis of influences on the initiation and maintenance of alcohol use behaviour evident from the data.[25 34 35] Thus, these theories were used as a theoretical lens, which helped to explain the data. We also explored whether social practice theory[36–38] might help to explain alcohol use behaviour as a collective social practice during adolescence. However, while a social practice perspective helped to explain the social context of alcohol use and the meanings associated with alcohol consumption, it did not fully explain data regarding peer influences and/or pressures, which were key themes in our study. We also note that we did

not specifically seek to examine the influence of social media in initial interviews, rather, the role and importance of social media emerged as a theme and was therefore explored in relation to social influences during data analysis.

## Public involvement

We did not involve young people in this study directly, however, the authors engaged with young people advisory groups (YPAGs) prior to commencing the programme of research in which this qualitative study was embedded. The overall aim of the programme of research was to develop an intervention to reduce excessive alcohol use and harm among young people. Together, this qualitative study, and engagement with YPAGs, aimed to inform the design and theoretical basis of an preventive intervention.

## RESULTS

The major themes that emerged from the data were: normalisation and the wider culture, motivations, peer influence, pressure among peers, social media and young people's drinking culture. Family influences were also a major theme, but we have not addressed them here, as we have previously reported similar findings elsewhere.[10]

### Normalisation of drinking and the wider alcohol culture

Young people described alcohol consumption as a normalised practice and an accepted part of the cultural context. Young people accepted drinking as part of life and stressed the importance of autonomy around people's decisions regarding alcohol consumption. Exposure to teen films and social media presented alcohol consumption as fun and 'cool' and a means of enhancing popularity, while lacking details of negative consequences. Young people therefore frequently came into adolescence with clear preformed positive attitudes towards drinking and an understanding of its meanings in the social world (online supplementary material S1, quotes 1 and 2).

> So it's just like you just grow up with like people around you like just drinking casually like with meals and stuff so no-one sees it as particularly dangerous… until like something actually goes wrong. Yeah. So you're kind of shielded from the actual dangers… (ID 17, F, drinker)

Reflecting the assimilation of wider cultural norms, perceptions of what constituted reasonable alcohol use involved levels of consumption far above levels at which drinking would become hazardous or harmful (see also online supplementary material S1, quote 3).

> INT: And how much do you think is ok would you say?
> RES: Not getting drunk like really drunk like you can't even walk. (ID 29, M, non-drinker)

Despite such misperceptions, participants (including those who did and did not drink alcohol) were aware of

the balance between consuming alcohol for enjoyment and negative consequences, evidencing a clear awareness of the potential for a range of negative consequences to ensue if limits were overstepped, rendering the activity pointless or creating difficulties for others (online supplementary material S1, quotes 4 and 6). In addition, those who abstained, while noting acceptance of alcohol use within social groups, highlighted avoidance of such consequences through the absence of drinking in their own world.

> I kind of fear the consequences of me drinking. And I also think about my family: if they were to find out they would just be like angry and annoyed at me. (ID 16, M, non-drinker)

### Motivations to drink and the initiation of alcohol use

Drinking became an integral practice in young people's social worlds by age 14 to 16. Young people most frequently described a shared meaning of alcohol consumption relating to collective fun, enjoyment and being cool. Drinking also improved confidence, popularity and engagement with friends while enabling young people to rebel, make memories and experiment, while providing a means of managing mental illness, avoiding reality, excusing behaviour or escaping stress (online supplementary material S2, quotes 1 and 2).

Satisfying the range of motivations to drink was straightforward since alcohol was both readily available and cheap. Some described initiation of drinking among peer groups via an influential individual, who was the first to provide or drink alcohol (often via parents), setting an expectation of drinking at subsequent parties.

> RES: I think first party was in October when it was like, oh there's going to be alcohol there. …And that was sort of an initiation. (LAUGHS) I don't know.
>
> INT: Yeah.
>
> RES: But it's sort of like the – yes it's after that party then it's sort of expected that there is at other ones, maybe. (ID 8, F, non-drinker)

### Young people's drinking and party culture

Drinking practice was rooted in house parties, which were viewed as a safer and more controlled setting than were outdoor settings. Parties' meanings were centred around the inherently social nature of the events, providing opportunities for socialising, shared experiences and a sense of togetherness with friends, as well as opportunity for drunkenness, which was a widely accepted norm. For many, alcohol made parties more enjoyable (online supplementary material S3, quote 1) and getting drunk provided an aim and purpose, while providing the means of looking popular, 'hard' or cool. Thus, widely held understandings, or aims, were to drink and 'get smashed' (online supplementary material S3, quotes 2 and 3), with the pace and extent of drunkenness often being a source of direct or indirect competition. Participants also described indirect pressures involving stigmatising insults linked to competence around intoxication.

> RES: …it's kind of like who can drink more and not get drunk, but it's like not a competition in that way, yeah.
>
> INT: So it's not a competition?
>
> RES: No, it's not a competition, but it feels like that because after you're a bit, oh, this guy's a lightweight, and that. (ID 37, M, non-drinker)

Thus, it was important to tread a line between appearing competent in tolerating alcohol to avoid stigmatising labelling as a 'lightweight' (online supplementary material S3, quote 4), but to remain in control to avoid ruining a party and/or require care from friends, which was associated with reputational risk and disgruntlement from friends. Those who often took on the role of looking after friends described feelings of reluctance and resentment around missing out on enjoyment, or concern around managing the effects of intoxication.

> The first feeling is panic and then you feel just disappointed and you're just angry. You feel… I feel a little bit angry but especially if they've done it before you know what you've done, you know how like what your limit is and yet you go out of your way to do it… (ID 15, F, moderate drinker)

The disparity between individual autonomy in decisions around intoxication and the requirement for help from others thereafter was also viewed negatively (online supplementary material S3, quote 5).

### Social influences on adolescent alcohol use
#### Peer influence

Social influences became paramount in mid-adolescence, with young people frequently reporting a desire to 'fit in' with friends. Those who drank earlier and to a greater extent acted as influential 'role models' for drinking (online supplementary S4.1, quote 1). Thus, young people implicitly understood the social significance of joining in and facilitating belonging in the social world to enhance social status or to avoid social sanctions.

> Kind of like 'cause everyone else is around you drinking, you kind of feel like you have to otherwise you don't really fit in um so that's what kind of happens to me most of the time. (ID 27, F, drinker)

Social influences also facilitated a rise in the prevalence of drinkers, thus enhancing proximal social influences and further contributing to the normalisation of alcohol consumption and the feedback cycle of conformity, with those who abstained thus feeling more isolated and dependent on non-drinking friends to diminish felt pressures.

Social influence led to perceived requirements to join in in some way, and those who didn't join in with the vibe of the party were a 'buzz kill' (online supplementary

material S4.1, quote 2). As such, a somewhat circular effect sustained the party culture, whereby the increasing prevalence of drinking strengthened group norms and social learning processes that generated a desire to drink to feel included, and which generated a social environment in which abstaining could be alien (see also online supplementary material S4.1, quote 3).

> I didn't intend to drinking alcohol last year… but it just kind of happens, like when you're invited to a party you just go and there's alcohol and you get, you get told there's going to be alcohol, and you just drink it to have a good time anyway 'cause everyone else is. (ID 39, M, previous, now infrequent drinker)

Those who didn't comply with norms described feelings of awkwardness, embarrassment or boredom or they avoided parties altogether, while others were incentivised to drink to avoid having to look after intoxicated friends (online supplementary material S4.1, quotes 4–6).

Notably, however, peer influences could have counter-effects, such that participants described alcohol consumption as pointless, expressing confusion around friends' drinking practices, while others reported aversion to role models of intoxication, which could strengthen views around abstinence (see also online supplementary material S4.1, quote 7).

> I wouldn't do it because I have seen what in the middle, I have seen people feeling sick, all in the bath, like going on the street, I was like no way I am not doing that, you see what the impacts are if you drink. You see what would happen and I don't want that at all. (ID 6, F, non-drinker)

Such individuals were less compliant, or non-compliant, with the group norm and often played a greater 'mum' role in supporting and helping others within the social group (online supplementary material S4.1, quote 8). Those who were non-compliant, acted in keeping with family values and messages, and were unaffected by group norms.

> I don't know what, what my mum would think if I become a drinker, so I – and I think she's kind of built it in me that it's, it's now in my moral code not to drink. (ID 12, M, non-drinker)

Individuals who abstained from drinking described the importance of friends with similar attitudes and behaviours around drinking and there was some evidence for peer selection in this group. Such friends provided support for their viewpoint and provided some protection from felt pressure or exclusion, enabling them to abstain and/or to enjoy social events more.

> …if I was on my own it would have been worse, because you're just looking around and everyone's drinking. Not that it would lead me to drinking, but it's still better that I had friends who just don't

do that at all: it made it much more fun. (ID 16, M, non-drinker)

Those who abstained frequently described strong interests in extracurricular activities, particularly sports, a strong religious faith and/or clear core values and aligned family viewpoints (which highlighted strong, clearly stated disapproval for alcohol use), which altered the meaning of alcohol use and reduced, or counteracted, the effects of social influences and norms (online supplementary material S4.1, quote 9). In addition, some of those who abstained reported family history of awareness of dependence or alcohol-related problems, which increased concern around the risks and negative consequences of alcohol consumption.

> I'm pretty sure I'm glad not to like I don't want to start doing it because then maybe like later life I will be addicted and I don't want to be. (ID20, M, non-drinker)

### The influence of social media

The use of social media was integral to adolescent drinking behaviour and practice, particularly use of Instagram or the multimedia messaging app Snapchat. Posting pictures and videos from parties reflected the meanings associated with alcohol use, thus portraying participants being cool and sociable, to enhance popularity and social status. The ubiquity and prominence of social media weaved into young people's lives a relatively consistent, positive view of alcohol consumption that could act as another source of social influence and a visual depiction of norms that could contribute to a felt pressure to drink (see also online supplementary material S4.2, quotes 1 and 2).

> Because in a friendship it's more like direct, like they're going to tell you that, but like, on a big scale like social media it kind of like tells you like that you should be drinking, like when you can, and the norm is to, and then I think you feel a bit pressured into it. (ID 37, M, non-drinker)

Social media used to demonstrate social capital, drinking 'competence' and maturity. Some were described as sipping drinks just to look right in pictures, while others aimed to depict their prowess in consumption.

> I've noticed that a lot in my year group and like they'll kind of just… even if they don't really want to drink it they'll kind of just hold it there and they'll just like sip and stuff so they can have it for photos. (ID 34, F, moderate drinker)

Nevertheless, views on the impact on social media were mixed. A minority expressed caution about posting images of themselves drinking owing to potential reputational risk. A minority also reported a view that drinking was unaffected by social media, although it was nevertheless described as a key component of drinking practice.

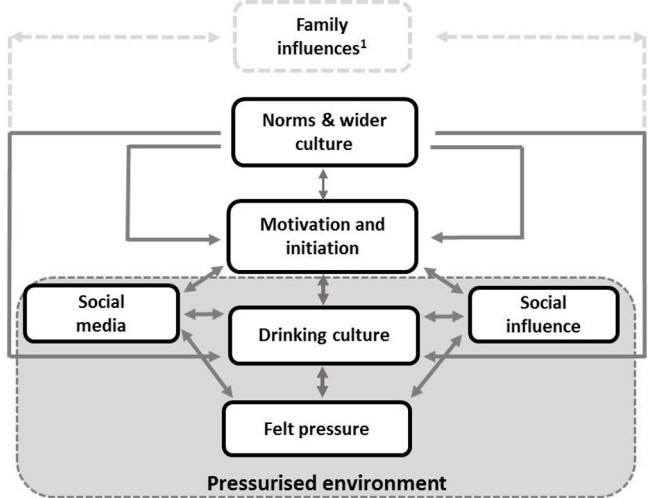

**Figure 1** Intersection of social and cultural influences on adolescent alcohol use and generation of a pressurised environment. [1]Family influences have been addressed elsewhere.

## The pressurised environment

The combination of social media, social influences and social learning, alongside the meanings associated with drinking (eg, belonging, popularity) and wider cultural influences, created a pressurised environment around drinking (figure 1). Thus, the very interest in fitting in or feeling a figure to join in with friends was felt to be subtly pressurising, while being the only person not drinking felt awkward and created unease, further contributing to a felt pressure.

> Because you see everyone's doing it then you felt… er like obliged to do it as well sometimes. Um I don't often, like, myself but you can definitely tell other people um it's like, oh everybody else is doing it so then I have to do it as well to fit in, it's like one of those sorts of things and also if your friends are drunk then it sort of becomes a thing where oh do you want mine you can have some, have some, and then you're like oh my gosh like do I accept, do I not. (ID 15, F, moderate drinker)

Branding of individuals as 'wimps' or awarding 'lad points' for intoxication associated with courage and humour, further created incentives and obligations to drink to excess (online supplementary material S5, quote 1). The inextricable link between drinking alcohol and socialising meant that resisting pressures or challenging friends risked losing credibility as an integral and fun group member, or risked jeopardising friendships (see also online supplementary material S5, quotes 2 and 3).

> INT: … and how does that make you feel if you are saying no?
> RES: Um if you're like receiving yeah then you can kind of feel like it makes you not want to be in that situation, it makes you just like feel oh I don't feel safe in this environment anymore cos now I'm doing

something that I don't want to do and you're just like what do I do, I can't deal with this situation and you don't want to not be friends with people who you're friends with, you don't want to break friendships over it but it's one of those things where you kind of if you don't like it then you have to say no and if your friends didn't accept that then it's like you have to sort that out with your friends yourself, that sort of thing. (ID 15, F, moderate drinker)

Pressure to drink was felt in a range of ways, including subtle and indirect pressure through offers of alcoholic drinks, the felt need to be fun and to fit in, or proximal social influence. However, participants also reported more direct pressure through goading or direct insults, leading some to describe feeling 'forced to drink' and others wishing to avoid appearing cowardly. Overall, the social environment was felt to be conducive to feeling obliged to drink, or being 'pushed in the direction' towards alcohol use (see also online supplementary material S5 quotes 4 and 5).

> … I used to think alcohol was really bad but I guess as like it becomes more talked about, you're kind of dragged into it.
> INT: Mm. It's interesting you say dragged into it.
> RES: Not dragged – I'd say like kind of um pushed in the direction. (ID 27, F, drinker)

Nevertheless, such felt pressures were often unclear to the friends in question who were drinking, who therefore pressurised others unknowingly. Drinkers did not consider their actions to be pressurising, rather a gesture to enable or encourage friends to join in (see also online supplementary material S5, quote 6).

> They just think they're saying it does – it's just a laugh like, if they're – they're kind of – they're almost trying to make their experience more enjoyable by saying if you do this… it will make it better, but I don't think that's the same kind of response for the person on the other end. (ID 33, M, drinker)

As outlined above, for non-drinkers, having friends with a similar viewpoint provided support and reduced felt pressure.

> … I don't know if I didn't have my friends or anyone to tell me not to drink, I'd probably be drinking by now. …So, I'm kind of happy that they're around. (ID 12, M, non-drinker)

## DISCUSSION

In this paper, we have demonstrated how the wider alcohol culture normalised alcohol use for young people, helping to generate perceptions of drinking as a social practice associated with being cool, mature and popular as well as an important means of enhancing enjoyment and social status, in support of evidence regarding primary social

and enhancement drinking motives in young adults.[39] The ubiquity of alcohol use generated normative social influences and opportunities for social learning around initiation and continuation of alcohol use. Together with embedded meanings and expectancies, the interaction between peer norms, social influences and the effects of social media generated a pressurised environment in which young people felt obliged to comply and join in.

The strengths of our study include the gender-balanced sample of young people from both schools and youth groups, including those who used alcohol and those who abstained. We also used one-to-one interviews, which provided space for young people to share their views around drinking, without concerns around disagreement or judgement from peers or friends. A small proportion of interviews were conducted with dyads with one triad, however, we did not find evidence that participants exaggerated or withheld views and perspectives in these pairs, and paired interviews were conducted when expressed as a preference by the participants. Nevertheless, our data were not triangulated, and participants were recruited from urban and suburban settings only, so our findings cannot necessarily be generalised to different geographical or demographic contexts. Interestingly, although parental messages and the off-putting effect of observing intoxicated role models were described by some moderate drinkers, our data did not provide evidence regarding factors that might be affecting downward trends in the prevalence of alcohol use among young people in England, and further qualitative and quantitative research will be needed to explore such trends. Lastly, while we have reported that social influences and pressures are a critical influence on young people's alcohol use, further research will be needed to explore exactly how such influences could be effectively addressed in public health interventions for young people.

The frequently reported desire to fit in with the group, and unease expressed in relation to abstinence, highlighted the central role of normative social influence in generating a collective practice in which intoxication was accepted and expected. Evidence suggests that both direct and indirect pathways of peer influences and pressures may exist, which can contribute to generating social influences (and can act together with peer selection[19 40]). Pathways can be direct, including offers of drinks, dares or goading, (fitting with social and enhancement motives for drinking)[41] and indirect, including modelling of peer drinking and/or contributions to the formation of norms, beliefs and expectancies,[42 43] (fitting with conformity motives for drinking)[39] which were also evident in our study. In addition, studies have reported an impact of being around peers who drink alcohol, either through proximity to social acquaintances who consume alcohol or through time spent with drinking peers,[44 45] highlighting the important impact of the rise in prevalence of alcohol use through adolescence. This was evident in our study since being around other drinkers and being immersed in a practice in which drinking was highly prevalent (contributing to descriptive norms) contributed to the felt obligation to drink.

Frequent reports by young people of needing to 'fit in' suggest that injunctive norms, or perceptions of peer approval of drinking, were also critical. Thus, our findings support those of quantitative studies, which have reported associations between both descriptive and injunctive norms and frequency and quantity of adolescence alcohol use and heavy episodic drinking.[22 46] Specifically, one study reported a greater impact of injunctive compared with descriptive norms, with the former associated with a range of measures of alcohol use as well as alcohol-related negative consequences.[22]

In addition to social norms and influences among peers, social media or SNS, played a role in providing a physical depiction of positive alcohol-related descriptive and injunctive norms, which was an integral part of the social practice of adolescent alcohol use. Thus, we have reported evidence to support a previous study describing 'intoxigenic digital identities' and 'intoxigenic digital spaces' that contribute to the normalisation of alcohol use.[47] Similarly, a qualitative study of young women's drinking cultures described the critical nature of SNS in facilitating displays of popularity, belonging, membership of the social network and 'friendship fun'.[32] Such media also bring about opportunities for misperceptions owing to the circulation of positive imagery, which can be interpreted at face value or as 'truth'.[48 49] Quantitative data have demonstrated an association between exposure to friends' alcohol-related SNS content and initiation of alcohol use and stronger pro-alcohol peer injunctive norms, highlighting how the provision of opportunities for observation, comparison and role-modelling of behaviours through SNS can impact on norm perception and behaviour.[24] In addition, others have reported an association between alcohol promotion through digital media and adolescent alcohol consumption.[50 51]

Taken together, therefore, evidence suggests the need for public health interventions to effectively address both peer norms and social influences during adolescence, including the effects of SNS, to enhance the likelihood of effective prevention. Furthermore, the evidence of felt pressure, resulting from the influences discussed above, suggests that many young people feel a lack of autonomy around alcohol consumption, and constraint around behavioural decisions within their social groups. Thus, in addition to addressing norm perception and normative social influence, there is a need for interventions to find ways of overcoming such constraints and fostering greater resistance to conformity. Indeed our sample included participants who reported abstaining from alcohol use, and who navigated the social world resisting influences of drinking peers, with critical support from non-drinking friends, the importance of which has been reported by others.[52] Thus, fostering greater resistance to conformity may require a strengthening of the voice and influence of those in the minority group (ie, those who abstain from drinking) as outlined in Moscovici's theory of minority

influence,[53 54] which may be increasingly feasible in light of reported downward trends in adolescent alcohol consumption[3] and thus an increase in the proportion of non-drinking peers. Moscovici also highlights how influence goes beyond shaping people to a system, but that it continually changes a social system, thus the influence of increasing proportions of abstainers, who in our study reported counter-effects of peer influence and aversion to role models of intoxication, may increasingly shift alcohol use downwards at a population level[25] as the benefits of the 'deviant' or non-conforming behaviour or perspective become evident. It has also been suggested that feelings of powerlessness during adolescence may be overcome through enhanced social status for young people, for instance through opportunities for creative activities and active engagement in communities.[55]

## CONCLUSIONS

Our study has demonstrated the effect of the interplay between cultural norms, social influences, social pressures and social media, particularly the critical role of normative social influence, in driving the escalation and normalisation of alcohol use in mid-adolescence. Our findings suggest that there is a need for a combination of approaches to effectively prevent excessive alcohol use among young people. Interventions in young people need to target normative social influence and peer norms to interrupt the rise in prevalence of excessive drinking, while aiming to enhance social status and the ability to challenge social pressures, to increase the likelihood of preventing alcohol-related harm.

**Contributors** GJM was responsible for recruitment, acquisition, analysis and interpretation of data and wrote the first draft of the manuscript. MH contributed to interpretation of data and finalisation of the manuscript. RC was responsible for conceptualisation of the study and contributed to interpretation of data and finalisation of the manuscript.

**Funding** GJM was supported by an NIHR post-doctoral fellowship award (PDF-2013-06-026). This report is independent research supported by the National Institute for Health Research (Post-Doctoral Fellowship, PDF-2013-06-026). The work was undertaken with the support of The Centre for the Development and Evaluation of Complex Interventions for Public Health Improvement (DECIPHer), a UKCRC Public Health Research Centre of Excellence. Joint funding (MR/KO232331/1) from the British Heart Foundation, Cancer Research UK, Economic and Social Research Council, Medical Research Council, the Welsh Government and the Wellcome Trust, under the auspices of the UK Clinical Research Collaboration, is gratefully acknowledged. RC is supported by the University of Bristol.

**Disclaimer** The views expressed in this publication are those of the author(s) and not necessarily those of the NHS, the National Institute for Health Research or the Department of Health and Social Care.

**Competing interests** RC is a scientific advisor to Evidence to Impact a not-for-profit company wholly owned by the Universities of Cardiff and Bristol which licences, quality-assures and supports the delivery of evidence-based public health promotion interventions. RC receives payment for this work. MH has received payment for unrelated activity from Gilead Sciences Inc, Bristol Myers-Squibb and Jannsen, UK.

**Patient consent for publication** Not required.

**Ethics approval** Ethical approval for the study was obtained from the University of Bristol Faculty of Health Sciences Ethics Committee (study 131443 (8201)/2306).

**Provenance and peer review** Not commissioned; externally peer reviewed.

**Data availability statement** No data are available.

**ORCID iDs**
Georgie J MacArthur http://orcid.org/0000-0003-2047-6519
Matthew Hickman http://orcid.org/0000-0001-9864-459X

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
