## [Reviewer comments · BMJ Open]

ARTICLE DETAILS

TITLE (PROVISIONAL)	A qualitative exploration of the intersection between social influences and cultural norms in relation to the development of alcohol use behaviour during adolescence.
AUTHORS	MacArthur, George; Hickman, Matthew; Campbell, Rona

VERSION 1 – REVIEW

REVIEWER	Gina Chrissy Martin Western University, London, Ontario, Canada
REVIEW RETURNED	16-Apr-2019

GENERAL COMMENTS	This paper presents some interesting qualitative work examining the role of peers in alcohol use among adolescents. There is much important insight to be drawn from the study. However, in its current form there is room for improvement that would add clarity to this work and make for a greater contribution to the literature. In the abstract- The context is not stated (urban/ suburban) or the country/region the study took place. The sentence “Social media weaved into...maturity” is unclear. The drinking status (abstainers and non-drinkers) should also be included up front. The statement regarding “escalation of excessive consumption” is unclear. Whose excessive consumption? The participants in the study?-but some were abstainers; the population?- but adolescent alcohol use in general is declining. The introduction- The introduction does not draw on any theoretical framework. There is also a lot of focus given to social media and the role it plays in adolescent drinking. However, it is not clear if this was something that was designed into the study at the onset or whether this was a theme that came out of the data later. Discussing the theoretical placement of the study is needed early on. Methods- It is stated that the aim of the study was to investigate the role of friendship but wider determinants are discussed. How were the interviews conducted and how did these wider determinants get brought up? This needs clarity. The authors state that this research is one part of a larger programme (in the public involvement section); however, it is not clear what the broader goals were and how this research fits into the project. Much more detail is needed in terms of project goals and design. How was the diversity of deprivation determined by a postcode? Was it a set of postcodes for a larger area that were diverse in terms of deprivation, or were schools selected from schools that had varying levels of deprivation based on postcode? Data analysis-
--

	This section is clear and gives a good overview of methods used. Results- A major strength of this study is that it included drinkers and abstainers, yet this isn't highlighted in the quotes (only M or F is reported) and only touched on in the analysis and discussion. This is one of the major contributions of this paper and the current analysis doesn't do it justice. Discussion- Much of the findings in this study are in line with previous work on drinking motivations. The results regarding motivations do not discuss the seminal work of drinking motivations by Cooper et al. The conclusion- "First, population – and family-level... cultural norms" is not supported by the data presented and discussed. Overall, there are some interesting findings in this work. Clarity is needed in terms of the goals of the project and whether the themes emerged out of the data or through the interviewer questions. A more theoretically grounded introduction that links to the analysis would improve the manuscript. Most importantly, given the declining rates of adolescent drinking, presenting the experience and views of those who abstain is an important element that is only touched on briefly in this work. Building on this would add depth and insight to the current knowledge base.
--	---

REVIEWER	Dr S J Scott Teesside University, UK
REVIEW RETURNED	13-Sep-2019

GENERAL COMMENTS	This is a well written and timely study that updates and adds to recently completed studies of drivers of adolescent alcohol use. I have some queries to be addressed by the authors who are to be complemented on an interesting piece of work:  1. It is unclear how the researchers obtained parental consent from participants recruited via youth groups, I wonder if there is space to clarify this as surely this was more complex than the process followed for participants recruited via schools? 2. Likewise, was parental consent obtained for all participants or only those under the age of 16? 3. Could the authors elaborate on how they felt data saturation was reached? At present, the paper simply states that it was reached not the reasoning and rationale for this. 4. Could the authors report the average interview length? 5. The manuscript appears to suggest that the data was solely analysed by the first author. Did any discussions take place between the whole team to explore emergent themes / data? It would be worthwhile to perhaps explore the work of Rose Barbour on coding and analysis of qualitative data here if the authors have not done so already? 6. It might be helpful, when reporting quotes, to include the age of the participant as well as their gender? With this in mind, did the authors note any age-related differences in influences or practices? 7. I'm not sure if the authors are working to a strict word count but it is really tricky going back and forth to supplementary material to look
---

	at related quotes. Could these simply be included in the body of the manuscript? 8. The authors mention the influence of a range of theoretical positions. Could they be more explicit about this in the manuscript? I can't see how data is linked to theory throughout the findings or in the discussion. 9. What reasons did abstainers provide for not drinking alcohol? This is not reported in findings despite being alluded to in methods.
--	---

VERSION 1 – AUTHOR RESPONSE

Reviewer 1

Abstract

The context is not stated (urban/ suburban) or the country/region the study took place. We have added in the methods section of the abstract that the interviews were conducted in ‘an urban centre in the West of England’

The sentence “Social media weaved into...maturity” is unclear. We have altered this sentence so that it now states: Social media presented to young people positive alcohol-associated depictions of social status, enjoyment and maturity.

The drinking status (abstainers and non-drinkers) should also be included up front. We have added detail about the drinking behaviour of participants.

The statement regarding “escalation of excessive consumption” is unclear. Whose excessive consumption? The participants in the study?-but some were abstainers; the population?- but adolescent alcohol use in general is declining. This sentence means to explain why the prevalence of alcohol use rises around age 15 i.e. mid-adolescence. We have altered this sentence to: driving the escalation in the prevalence of excessive consumption at this stage.

Introduction

The introduction does not draw on any theoretical framework. There is also a lot of focus given to social media and the role it plays in adolescent drinking. However, it is not clear if this was something that was designed into the study at the onset or whether this was a theme that came out of the data later. Discussing the theoretical placement of the study is needed early on.

We are grateful for this point and have included additional clarification towards the end of the introduction, stating:

In this paper, we report findings of a qualitative study that used theories of social influences, group identity and social norms as the theoretical lens through which to explore the social, cultural and behavioural drivers of alcohol consumption and the nature of drinking culture in mid-adolescence.

Since we also mention the theoretical considerations made during analysis in the methods section, we consider that mention of the focus on social media is best placed here. As such, we have added a point in the data analysis section of the Methods stating:

We also note that we did not specifically seek to examine the influence of social media in initial

interviews, rather, the role and importance of social media emerged as a theme and was therefore explored in relation to social influences during data analysis.

Methods

It is stated that the aim of the study was to investigate the role of friendship but wider determinants are discussed. How were the interviews conducted and how did these wider determinants get brought up? This needs clarity.

Interviews were semi-structured and were conducted using a topic guide. Since we used a semi-structured approach, there was flexibility for young people to introduce and explore issues that were meaningful to them, hence why wider determinants were raised and therefore incorporated into analysis of the data.

We have amended the first paragraph of the methods to clarify, so that it now reads:

While the initial aim of the study was to investigate the role of friendships in relation to drinking behaviour, we report our findings regarding this and a wider range of determinants of behaviour, which reflects additional topics pertinent to young people that were raised by them and discussed in the interviews and which thereby feature in the data analysis.

The authors state that this research is one part of a larger programme (in the public involvement section); however, it is not clear what the broader goals were and how this research fits into the project. Much more detail is needed in terms of project goals and design.

The qualitative study was conducted as part of a wider programme of research conducted to inform development of an intervention to reduce excessive alcohol use and related harms among young people. We have added additional content in this paragraph, now stating:

We did not involve young people in this study directly, however, the authors engaged with young people advisory groups (YPAGs) prior to commencing the programme of research in which this qualitative study is embedded. The overall aim of the programme of research is to develop an intervention to reduce excessive alcohol use and harm among young people and this qualitative study, and engagement of YPAGs aims to inform the design and theoretical basis of a preventive intervention.

How was the diversity of deprivation determined by a postcode? Was it a set of postcodes for a larger area that were diverse in terms of deprivation, or were schools selected from schools that had varying levels of deprivation based on postcode?

We realise that there is a lack of clarity around this point and are glad for the opportunity to correct this point in the manuscript. The index of multiple deprivation score for the ward within which the school was located was identified. Schools were grouped by IMD score and a random number generator used to select a school from each group to be contacted regarding participation. The final sample of four schools represented wards with varied ward-level IMD scores.

Data analysis-

This section is clear and gives a good overview of methods used.

Results-

A major strength of this study is that it included drinkers and abstainers, yet this isn't highlighted in the quotes (only M or F is reported) and only touched on in the analysis and discussion. This is one of the major contributions of this paper and the current analysis doesn't do it justice.

We have added by the quotes whether the participant was a non-drinker, light or moderate drinker or drinker, based on self-report of behaviour during the interview.

We have also added content to the results section and discussion relating to the views of abstainers. However, it is noteworthy that those who chose not to drink still discussed feelings of pressure and experiences of being among friends who did consume alcohol and so this is reflected in comments and quotes in the paper.

Discussion-

Much of the findings in this study are in line with previous work on drinking motivations. The results regarding motivations do not discuss the seminal work of drinking motivations by Cooper et al.

Our findings regarding perceptions of alcohol as being fun, cool and enhancing social enjoyment indeed support the work of Cooper et al, and we have noted this at the start of the discussion.

The conclusion-

“First, population – and family-level... cultural norms” is not supported by the data presented and discussed.

We have removed this sentence from the conclusions.

Overall, there are some interesting findings in this work. Clarity is needed in terms of the goals of the project and whether the themes emerged out of the data or through the interviewer questions. A more theoretically grounded introduction that links to the analysis would improve the manuscript.

We have included additional content regarding the aims and theoretical basis of our work, as well as how the themes emerged, as noted above. We have reduced the amount of content regarding social media in the introduction and have added detail about the theories relevant to our data analyses.

Most importantly, given the declining rates of adolescent drinking, presenting the experience and views of those who abstain is an important element that is only touched on briefly in this work. Building on this would add depth and insight to the current knowledge base.

We have strengthened content regarding those who reported abstaining from alcohol use. We note that the manuscript includes content regarding their motives to abstain and their views and perspectives around others who drink, as well as the importance of other non-drinking friends, which we consider adds to the evidence base around adolescent alcohol use.

Reviewer 2

This is a well written and timely study that updates and adds to recently completed studies of drivers of adolescent alcohol use. I have some queries to be addressed by the authors who are to be complemented on an interesting piece of work:

1. It is unclear how the researchers obtained parental consent from participants recruited via youth groups, I wonder if there is space to clarify this as surely this was more complex than the process followed for participants recruited via schools?

We have added a point in the methods section, noting that youth group leaders disseminated information about the study and study materials, and also arranged interviews and the focus group with participants.

2. Likewise, was parental consent obtained for all participants or only those under the age of 16?

We have clarified that parental consent was obtained for those aged under 16.

3. Could the authors elaborate on how they felt data saturation was reached? At present, the paper simply states that it was reached not the reasoning and rationale for this.

We have added content in the manuscript so that it now reads:

The number of participants recruited was determined by the point at which saturation was reached i.e. when no new themes or perspectives were emerging in the interviews.

4. Could the authors report the average interview length?

The average length of the interviews was 39 minutes and we have added this to the manuscript.

5. The manuscript appears to suggest that the data was solely analysed by the first author. Did any discussions take place between the whole team to explore emergent themes / data? It would be worthwhile to perhaps explore the work of Rose Barbour on coding and analysis of qualitative data here if the authors have not done so already?

The first author analysed the data, but emerging themes and theoretical bases for the findings were discussed with the senior author on the paper. We have noted this in the methods section so that it now reads:

Emerging themes and concepts, and the theoretical basis for analysis, was discussed with the last author.

We are grateful for the point about Rose Barbour's work.

6. It might be helpful, when reporting quotes, to include the age of the participant as well as their gender? With this in mind, did the authors note any age-related differences in influences or practices? The majority of participants were in year 10 and therefore aged 14-15 years therefore we cannot comment accurately on age-related differences. However, participants included individuals who had learnt from experiences and thus changed their behaviour, or who had started to drink, as well as those who abstained. This was not necessarily dependent on age, but on age at initiation, friendship group etc.

7. I'm not sure if the authors are working to a strict word count but it is really tricky going back and forth to supplementary material to look at related quotes. Could these simply be included in the body of the manuscript?

We have adhered to the word count of the journal so are reluctant to include the additional quotes in the body of the manuscript as doing this would amount to a substantial increase over the recommended word count.

8. The authors mention the influence of a range of theoretical positions. Could they be more explicit about this in the manuscript? I can't see how data is linked to theory throughout the findings or in the

discussion.

By exploring sociological theory around social norms and social influences, we gained a stronger understanding of the processes by which behaviour was influenced by the actions of others, and injunctive and descriptive norms; and thus how the data were explained by these theories. This enabled framing of findings within these theoretical constructs. We investigated whether the data could be explained by social practice theory, but since it did not fully explain the data, we did not consider this theoretical position further during analysis of the data. We have added content into the methods section.

To address this point, and points made by Reviewer 1, we have added content into the introduction to highlight the theoretical basis for the paper and our discussion highlights the importance of social influences and social norms, as well as detailing the evidence base of how peer influences and norms affect alcohol consumption. We consider that this strengthens the link between theory and the data.

9. What reasons did abstainers provide for not drinking alcohol? This is not reported in findings despite being alluded to in methods.

We have added content into the results section (peer influence section, page 17) to explain this and have included additional content and quotes from abstainers in the results section.

VERSION 2 – REVIEW

REVIEWER	Steph Scott Teesside University UK
REVIEW RETURNED	31-Jan-2020
GENERAL COMMENTS	Thank you for your detailed and thorough response to queries. I have no further questions, and the authors are to be congratulated on a strong manuscript.